# Viral Metagenomics Revealed Sendai Virus and Coronavirus Infection of Malayan Pangolins (*Manis javanica*)

**DOI:** 10.3390/v11110979

**Published:** 2019-10-24

**Authors:** Ping Liu, Wu Chen, Jin-Ping Chen

**Affiliations:** 1Guangdong Key Laboratory of Animal Conservation and Resource Utilization, Guangdong Public Laboratory of Wild Animal Conservation and Utilization, Guangdong Institute of Applied Biological Resources, Guangzhou 510260, China; pingliu0330@126.com; 2Guangzhou Zoo, Guangzhou 510230, China; chenwu-01@163.com

**Keywords:** virome, *Manis javanica*, Sendai virus, Coronavirus, molecular epidemiology

## Abstract

Pangolins are endangered animals in urgent need of protection. Identifying and cataloguing the viruses carried by pangolins is a logical approach to evaluate the range of potential pathogens and help with conservation. This study provides insight into viral communities of Malayan Pangolins (*Manis javanica*) as well as the molecular epidemiology of dominant pathogenic viruses between Malayan Pangolin and other hosts. A total of 62,508 *de novo* assembled contigs were constructed, and a BLAST search revealed 3600 ones (≥300 nt) were related to viral sequences, of which 68 contigs had a high level of sequence similarity to known viruses, while dominant viruses were the Sendai virus and Coronavirus. This is the first report on the viral diversity of pangolins, expanding our understanding of the virome in endangered species, and providing insight into the overall diversity of viruses that may be capable of directly or indirectly crossing over into other mammals.

## 1. Introduction

The Malayan pangolin (*Manis javanica*), a representative mammal species of the order Pholidota, is one of the only eight pangolin species worldwide. Four of them are from Asia (*M. javanica*, *M. pentadactyla*, *M. crassicaudata* and *M. culionensis*), whereas another four from Africa (*M. tricuspis*, *M. tetradactyla*, *M. gigantea* and *M. temminckii*) [1]. Unlike other placental mammals, the skin of pangolins is covered by large and overlapping keratinized scales [2]. Because of the huge demand for their meat as a delicacy and their scales for use in traditional medicines, pangolins are the most poached and trafficked mammal in the world. That is why all the eight pangolin species are included in the Convention on International Trade in Endangered Species of Wild Fauna and Flora (CITES). Concerted efforts have been made to conserve and rescue these species in captivity in China because of their threatened status and continuing decline of the population size in the wild. At the same time, poor health condition and low immunity are also important problems for the rescue of pangolins. A previous study reported a complete genome sequence of Parainfluenza Virus 5 (PIV5) from a Sunda Pangolin (the same as Malayan Pangolin) in China, which further broadens the PIV5 infection host spectrum [3], implicating that pangolins are not only confronted with the potential of great harm from humans, but are also facing the risk of infectious diseases. Recently, a large number of viral metagenomic studies have found pathogenic viruses carried by human, pig, cow, bat, cat, horse, chicken and other animals [4,5,6,7,8,9,10], some of which successfully isolated new virus strains. However, we still know little about the diseases and their etiologies of rare and threatened terrestrial vertebrate such as pangolins.

Viruses are infectious agents that replicate only inside living cells and have the ability to infect a variety of hosts [11]. There has been a lot of discussion within the virology community regarding the best method to determine viral infectivity, pathogenicity, and effects on the host microbiome. Virologists use a variety of methods to gain understanding of infection, replication, pathogenicity, and, more recently, the evolution of the viral genome. Unbiased sequencing of nucleic acids from environmental samples has great potential for the discovery and identification of diverse microorganisms [12,13,14,15]. We know this technique as metagenomics, or random, agnostic or shotgun high-throughput sequencing. In theory, metagenomics techniques enable the identification and genomic characterization of all microorganisms present in a sample with a generic laboratory procedure [16]. The approach has gained popularity with the introduction of next-generation sequencing (NGS) methods that provide more data in less time at a lower cost than previous sequencing techniques. While initially mainly applied to the analysis of the bacterial diversity, modifications in sample preparation protocols allowed characterization of viral genomes as well. Researchers have seized the opportunity to expand our knowledge in the fields of virus discovery and biodiversity characterization [12,13,15,17].

The Guangdong Wildlife Rescue Center received 21 live Malayan pangolins from the Anti-smuggling Customs Bureau on 24 March 2019; most individuals, including adults and subadults, were in poor health, and their bodies were covered with skin eruptions. All these Malayan pangolins were rescued by the Guangdong Wildlife Rescue Center, however, 16 died after extensive rescue efforts. Most of the dead pangolins had a swollen lung which contained a frothy liquid, as well as the symptom of pulmonary fibrosis, and in the minority of the dead ones, we observed hepatomegaly and splenomegaly. We collected 21 organ samples of lung, lymph, and spleen with obvious symptoms from 11 dead Malayan pangolins to uncover the virus diversity and molecular epidemiology of potential etiologies of viruses based on a viral metagenomic study. This study will be beneficial to pangolin disease research and subsequent rescue operation.

## 2. Materials and Methods

### 2.1. Ethics Statement

The study design was approved by the ethics committee for animal experiments at the Guangdong Institute of Applied Biological Resources (reference number: GIABR20170720, 20 July 2017) and followed basic principles outlined by this committee.

### 2.2. Library Preparation and Sequencing

In our study, organ samples of lung, lymph and spleen were collected from dead Malayan pangolins at the Guangdong Wildlife Rescue Center. Preparation of viral-like particles followed a previous published paper [18]. Total nucleic acid was extracted from viral-like particles using a MagPure Viral DNA/RNA Mini LQ Kit (R6662-02; Magen, Guangzhou, China). Double-stranded cDNA was synthesized by reverse transcription from single-stranded and double-stranded RNA viral nucleic acids using REPLI-g Cell WGA & WTA Kit (150052; Qiagen, Hilden, Germany), while single-stranded DNA viral nucleic acids were converted to double-stranded DNA and purified by a REPLI-g Cell WGA & WTA Kit (150052; Qiagen, Hilden, Germany). Amplified DNA was randomly sheared by ultrasound sonication (Covaris M220) to produce fragments of ≤ 800 bp, and sticky ends repaired and adapters added using T4 DNA polymerase (M4211, Promega, Madison, WI, USA), Klenow DNA Polymerase (KP810250, Epicentre), and T4 polynucleotide kinase (EK0031, Thermo scientific-fermentas, Glen Burnie, MD, USA). Fragments of approximately 350 bp were collected by beads after electrophoresis. After amplification, libraries were pooled and subjected to 150 bp paired-end sequencing using the Novaseq 6000 platform (Illumina, San Diego, CA, USA). High-throughput sequencing was conducted by the Magigene Company (Guangzhou, China). The data supporting this study are openly available on the NCBI sequence read archive (SRA) under Bio Project PRJNA573298.

### 2.3. Raw Read Filtering and Rapid Identification of Virus Species

#### 2.3.1. Quality Control

As raw sequencing reads always include some low-quality data, it is necessary to perform processing to improve the accuracy of reads for follow-up analyses. To this end, we used SOAPnuke version 1.5.6 [19] to remove adapter sequences and reads (i) with more than 5% Ns; (ii) those with 20% base quality values less than 20; (iii) those arising from PCR duplications; as well as (iv) those with a polyA sequence.

#### 2.3.2. Remove Host Contamination

To avoid the confusion cause by ribosomes and host sequences, all clean reads that passed quality control were mapped to the ribosomal database (silva) and the host reference genome of *M. javanica* (NCBI Project ID: PRJNA256023) utilizing BWA version 0.7.17 [20]; only the unmapped sequences were used in subsequent analysis.

#### 2.3.3. Rapid Identification of Virus Species

Clean reads without ribosomes and host sequences were mapped to an in-house virus reference data separate from the GenBank non-redundant nucleotide (NT) database to primarily identify virus reads. According to the NCBI taxonomy database annotation information, reads were classified into different virus families. To improve the accuracy, we removed the alignment results with a coverage below 5 reads.

### 2.4. Read Assembly and Species Identification

Clean reads were *de novo* assembled using MEGAHIT version 1.0 [21]. BWA version 0.7.17 [20] was used to align clean reads to assembled contigs. A host sequence was determined based on BLAST version 2.7.1 and was removed by satisfying one of the following conditions: (1) Length of matched area ≥ 500 bp, alignment similarity ≥ 90%; (2) Length of matched area accounts for more than 80% of the total length of contigs, and alignment similarity ≥ 90%. Then, Cdhit version 4.6 [22] was used to cluster the assembled virus contigs from each Malayan pangolin sample. Contigs were then classified by BLASTx against the NT database using alignment similarity ≥ 80%, length of matched area ≥ 500 bp and e-value ≤ 10^−5^. Contigs with significant BLASTx hits were confirmed as virus sequences.

### 2.5. Phylogenetic Analysis

Whole genome sequences of virus strains, the same species as the dominant viruses in Malayan pangolins, from different hosts were downloaded from ViPR database (https://www.viprbrc.org/brc/home.spg?decorator=vipr). Virus sequences from Malayan pangolins and other hosts were aligned using MAFFT version 7.427 [23] with the auto alignment strategy. The best substitution models, as well as maximum likelihood (ML) trees were then evaluated with the iqtree version 1.6.9 [24] with 1000 bootstrap replicates. Then, all the ML trees were visualized and exported as vector diagrams with FigTree version 1.4.3 (http://tree.bio.ed.ac.uk/software/figtree/).

## 3. Results

### 3.1. Viral Metagenomics

A total of 21 organ samples of lung, lymph and spleen from 11 dead Malayan pangolins that could not be rescued by the Guangdong Wildlife Rescue Center were used to reveal viral diversity of pangolins. Viral nucleic acids were deep sequenced and then we obtained a total of 227.32 GB data (757,729,773 valid reads, 150 bp in length). In total, 233,587 reads were best matched with viral proteins available in the NCBI NR database (~0.03% of the total sequence reads). The number of viral-associated reads in each sample varied from 2856 to 78,052 (Table 1). In the aggregate, 28 families of viruses were parsed (Appendix A). The most widely distributed virus families were *Herpesviridae* and *Paramyxoviridae*, and the diverse reads related to these families occupied ~85% of the total viral sequence reads (Figure 1).

Contig sequences were then generated by *de novo* assembly using MEGAHIT version 1.0 [21], generating 62,508 unique contigs with a max. length of 13,503 bp (Table 2, Appendix A). A taxonomic assignment of these contigs was performed on the basis of BLAST analysis. At this stage, 68 contigs were confirmed for virus species, accounting for about 0.1% of the total number of contigs (Table 2). An assignment of these contigs to different types of viral genomes identified 20.59% DNA viruses and 79.41% RNA viruses, among which 14.71% were assigned to Phages. Another 3532 contigs were suspected to be assigned to virus species (Appendix A). DNA viruses accounted for 66.53% while RNA viruses accounted for 33.47%, and 37.06% of these contigs were assigned to Phages. For all the unique contigs, the top 30 ones with the most reads abundance were assigned to families *Paramyxoviridae*, *Flaviviridae* and *Caudovirales* (Figure 2).

### 3.2. Sendai Virus

Sendai virus was identified in 6 of the 11 Malayan pangolin individuals, which was the common identified virus. For several of these pangolin samples, larger Sendai virus contigs were produced (Table 2). In one case, a contig of 13,232 base pairs isolated from the lung tissue of individual 19 was identified, which is about 86% of the whole genome sequence length (15,384). This contig showed relatively high sequence identity (89.76%) to the whole genome sequence of a Sendai virus strain isolated from humans (GenBank accession: AB005795). The length of other contigs conformed as Sendai virus was in the range from 608 to 7027 bp (Table 2).

Whole genome sequences of Sendai virus from human beings, mouse and monkey were downloaded from the ViPR database (https://www.viprbrc.org/brc/home.spg?decorator=vipr). After sequence alignment conducted by MAFFT version 7.427 (Katoh & Standley, 2013), the best substitution model analyzed by iqtree was GTR+F+I. Then phylogenetic analysis revealed the closest relationship between the 13,232 bp length contig from Malayan pangolin and Sendai virus strains isolated from humans (AB005795.1), but distant from strains isolated from the mouse (Figure 3). Then, we generated the phylogenetic relationships of each gene sequence. Six trees had slight differences, but the genetic distance between the Sendai virus from Malayan pangolin and humans (AB005795.1) was the closest (Figure 4); the same as the relationship between them generated based on whole genome sequences.

### 3.3. Coronavirus

One or several members of the *Coronaviridae* families were identified in 2 out of the 11 *M. javanica* individuals (individual 07 and 08). For several of these pangolin samples, larger contigs were produced, and the length ranged from 503 to 2330 bp (Table 3). Though there was high species variety of Coronavirus detected, SARS-CoV was the most widely distributed (Table 3). Whole genome sequences of strains belonging to four genera (*Alphacoronavirus*, *Betacoronavirus*, *Gammacoronavirus*, and *Deltacoronaviruses*) isolated from different hosts were downloaded from the ViPR database (https://www.viprbrc.org/brc/home.spg?decorator=vipr). Together with 16 contigs confirmed as Coronavirus in this study, all the sequences were aligned utilizing MAFFT version 7.427 (Katoh & Standley, 2013). The best substitution model analyzed by iqtree was GTR+F+R7, and the phylogenetic analysis therefore showed multiple relationships between Coronavirus contigs and the four Coronavirus genera (Figure 5).

## 4. Discussion

Pangolins are important wildlife resources in imminent danger of extinction. Great efforts have been made to rescue trafficked pangolins; however, most of the pangolin individuals intercepted by customs were in a poor health condition, and then dead in a few days. Investigating the potential pathogens carried by pangolins may help to rescue them. Our viral metagenomics analysis revealed a high diversity of viruses carried by dead Malayan pangolins. The Sendai virus and Coronaviruses were dominant virus species conformed by assembled contigs, which might have some relationship with the death of Malayan pangolins. Recently, the prediction of viral zoonosis epidemics has become a major public health issue. A profound understanding of the viral population in key animal species acting as reservoirs represents an important step towards this goal. Bats are natural hosts for a large variety of zoonotic viruses. In a recently study, up to 47 different virus families were detected from bat fecal samples [25]. Over 130 virus species have been detected in bats as of 2017 [26], including several emergent human pathogens [27,28,29,30,31,32,33,34,35,36,37,38]. For domesticated animals, virome analysis between sick and health ones could help to find out the pathogens or virus diversity [8,9,39,40,41]. Our study showed that viral metagenomics analysis could also work in revealing viral diversity and potential pathogens of rare and threatened terrestrial vertebrates such as pangolins.

The Sendai virus was the most widely distributed pathogens in 11 dead Malayan pangolins, which was one of the potential causes of their death. The whole genome and individual gene phylogenies for Sendai virus sequences assembled in this study all showed that the Sendai virus from Malayan pangolin had the closest relationship with the strain isolated from humans (AB005795.1), which strongly suggests the possibility that the Sendai virus is transmitted between pangolins and humans. Sendai virus is a member of the paramyxovirus subfamily *Paramyxovirinae*, genus Respirovirus, members of which primarily infect mammals. The scientific community considers the Sendai virus as the archetype organism of the *Paramyxoviridae* family because most of the basic biochemical, molecular and biologic properties of the whole family were derived from its own characteristics [42]. Sendai virus-associated disease has a worldwide distribution and has been found in mouse colonies in Asia [43], North America [44] and Europe [45], and is responsible for a highly transmissible respiratory tract infection in mice, hamsters, guinea pigs, rats, and occasionally pigs and bats [46,47], with infection through both air and direct contact routes. Epizootic infections of mice are usually associated with a high mortality rate, while enzootic disease patterns suggest that the virus is latent and can be cleared over the course of a year. This is the first report of a wild pangolin dying possibly due to Sendai virus infection, which further broadens the Sendai virus infection host spectrum. Because of the lack of healthy individuals as a control, we could not figure out whether the Sendai virus carried by pangolins was caused by infection from other hosts or was inherited.

Besides the Sendai virus, Coronaviruses were also detected as potential pathogens of Malayan pangolins. The phylogeny of Coronavirus sequences assembled and strains from four Coronavirus genera demonstrated complex genetic relationships and high species diversity of the Coronavirus in Malayan pangolins. Coronaviruses can cause a variety of severe diseases including gastroenteritis and respiratory tract diseases, and have been identified in mice, rats, chickens, turkeys, swine, dogs, cats, rabbits, horses, cattle and humans [48,49]. Sometimes, but not often, a coronavirus can infect both animals and humans. Human coronaviruses were first described in the 1960s for patients with the common cold. Since then, more have been discovered, including those that cause severe acute respiratory syndrome (SARS) and Middle East respiratory syndrome (MERS), two pathogens that can cause fatal respiratory disease in humans [50,51]. It was recently discovered that dromedary camels in Saudi Arabia harbor three different human coronaviruses species, including a dominant MERS HCoV lineage that was responsible for the outbreaks in the Middle East and South Korea during 2015 [52]. The detection of different types of SARS-CoV in this study may also be related to the death of the Malayan pangolins. Considering the outbreak of SARS which was transmitted by masked palm civet from the natural reservoir of bats [29,53,54], Malayan pangolins could be another host with the potential of transmitting the SARS coronavirus to humans. As a consequence, the viral metagenomic study of Malayan pangolin is meaningful both for the conservation of rare wild animals and public health.

## 5. Conclusions

We found high viral diversity of dead Malayan pangolins, and the Sendai virus and Coronavirus may be the dominant pathogens responsible for their death. The Sendai virus showed a close relationship between the Malayan pangolin and the strain isolated from humans, whereas Coronavirus sequences showed a high species diversity. Further investigations are required to compare the incidence of these viruses in healthy and diseased pangolin individuals in order to better elucidate their pathogenic role. To date, this is the first metagenomic study of virus diversity in pangolins in China. This study expands our understanding of the viral diversity in endangered species and the capability of directly or indirectly crossing over into other mammals.

## Figures and Tables

**Figure 1 viruses-11-00979-f001:**
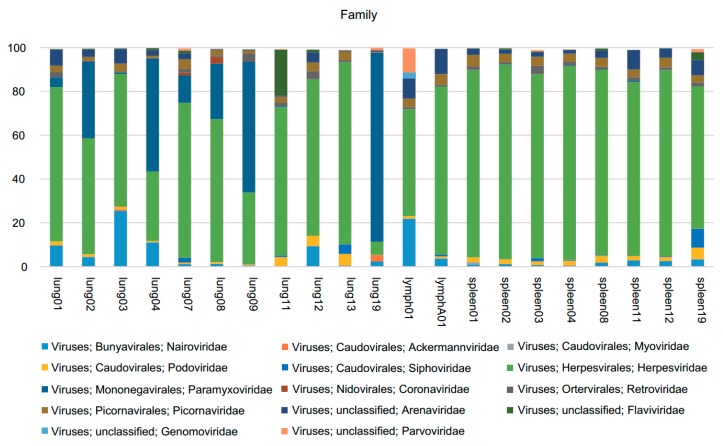
The percentage of sequences related to the most abundant viral families among all virus reads, indicated in the same colors for each main viral category. Taxonomic classification of viruses is consistent across samples. Samples are characterized according to the number of sequences from each sample classified by taxonomic family. Virus families are indicated by the color code on the bottom.

**Figure 2 viruses-11-00979-f002:**
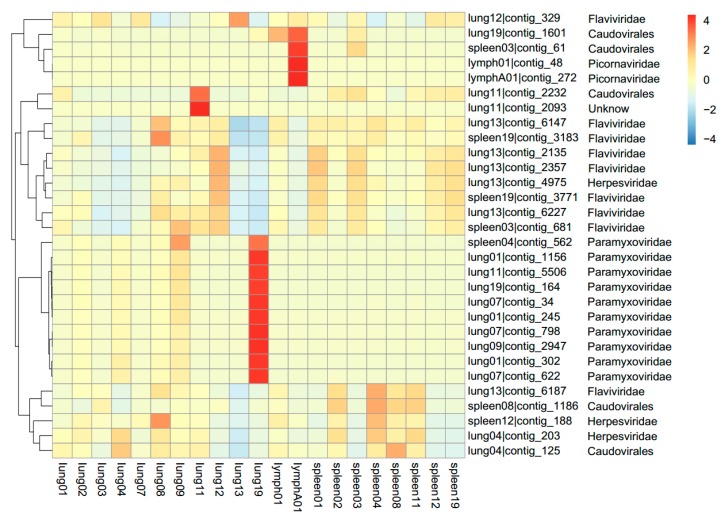
Heatmap of contigs with the top 30 abundance of sequence reads in each sample. The pangolin samples are listed below the heatmap. Information of contigs and the virus families they belong to is provided in the right text column. The boxes colored from blue to red represent the abundance of virus reads aligned to each contig.

**Figure 3 viruses-11-00979-f003:**
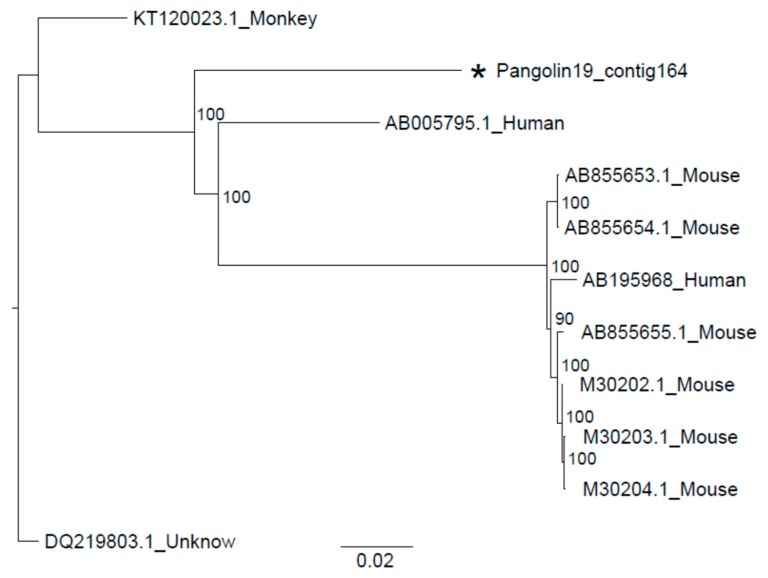
The phylogenetic tree of the Sendai virus from Malayan pangolin and other hosts. The analysis was inferred using the Maximum Likelihood method based on iqtree [24]. Branch bootstrap values are shown and were based on 1000 replicates. The black star indicates a contig of the Sendai virus from *M. javanica* in 2019.

**Figure 4 viruses-11-00979-f004:**
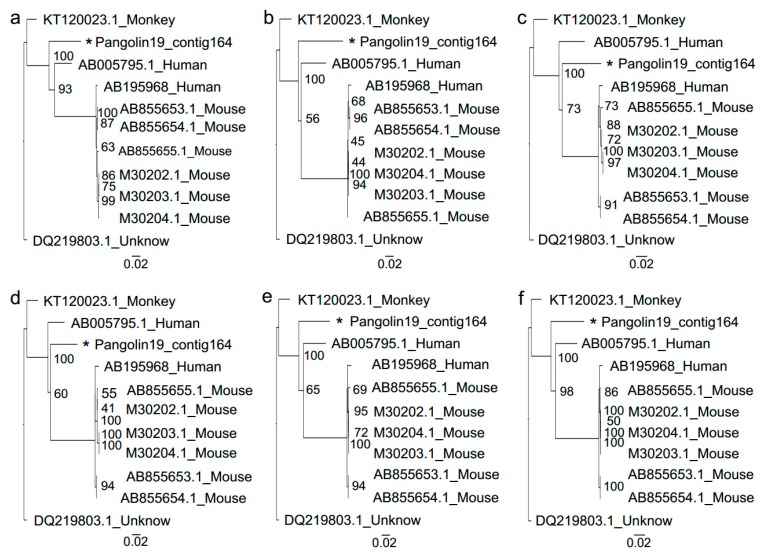
The phylogenetic tree of each gene of the Sendai virus from Malayan pangolin and other hosts. The analysis was inferred using the Maximum Likelihood method based on iqtree [24]. Branch bootstrap values are shown and were based on 1000 replicates. The black star indicates a contig of the Sendai virus from *M. javanica* in 2019.

**Figure 5 viruses-11-00979-f005:**
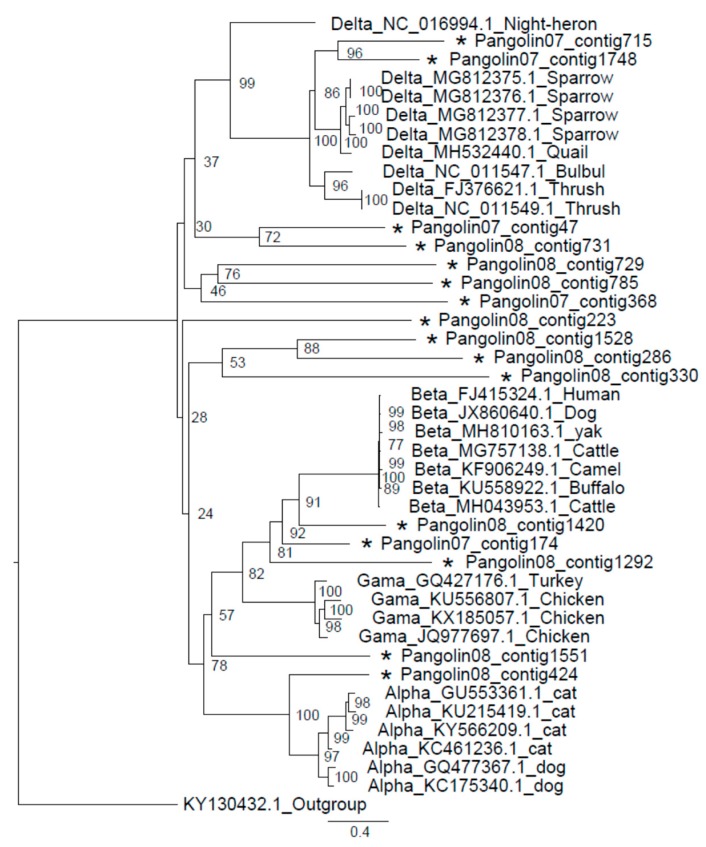
The phylogenetic tree of Conronavirus from Malayan pangolin and other hosts. The analysis was inferred using the Maximum Likelihood method based on iqtree [24]. Branch bootstrap values are shown and were based on 1000 replicates. The black star indicates a contig of the Sendai virus from *M. javanica* in 2019.

**Table 1 viruses-11-00979-t001:** Overview of reads and contig sequences of lung, lymph, and spleen tissues from 11 dead Malayan pangolins.

Sample ID	Raw Reads (PE)	Number of Reads Remaining after Filtering (%)				Assembly Data on Filtered Reads				
		Clean Reads (PE)	Rm. rRNA Clean (PE)	Rm. Host Clean (PE)	Virus Reads (PE)	Total No.	Max Len.	Min Len.	N50	GC (%)
lung01	53,970,685	22,900,426 (42.43)	13,929,751 (60.83)	13,784,503(60.19)	8945 (0.04)	2395	7054	300	471	51.08
lung02	39,738,679	16,573,376 (41.71)	10,760,690 (64.93)	10,580,567 (63.84)	10,242 (0.06)	3828	7638	300	534	51.02
lung03	29,005,761	12,967,281 (44.71)	7,511,236 (57.92)	7,427,749 (57.28)	7456 (0.06)	3380	5192	300	490	47.60
lung04	32,420,343	13,527,964 (41.73)	8,156,824 (60.30)	7,838,436 (57.94)	15,539 (0.11)	6047	7392	300	559	50.72
lung07	44,500,928	19,045,923 (42.80)	12,466,935 (65.46)	12,339,084 (64.79)	6056 (0.03)	2539	2541	300	429	47.50
lung08	39,624,368	16,414,925 (41.43)	10,655,020 (64.91)	10,555,677 (64.31)	9139 (0.06)	2196	6969	300	514	50.72
lung09	42,219,253	18,067,615 (42.79)	11,552,994 (63.94)	11,442,175 (63.33)	13,146 (0.07)	4903	13,503	300	623	46.97
lung11	52,714,790	22,220,187 (42.15)	15,402,765 (69.32)	14,227,635 (64.03)	11,877 (0.05)	9668	4560	300	394	49.82
lung12	17,630,092	9,275,501 (52.61)	5,425,644 (58.49)	5,368,963 (57.88)	2856 (0.03)	638	2866	300	422	52.31
lung13	25,571,230	16,491,648 (64.49)	14,588,679 (88.46)	10,591,839 (64.23)	7447 (0.05)	7557	8164	300	495	62.37
lung19	39,314,715	19,986,780 (50.84)	9,028,100 (45.17)	8,889,856 (44.48)	78,052 (0.39)	2,469	13,232	300	509	51.16
lymph01	40,842,452	18,903,834 (46.28)	11,243,800 (59.48)	11,117,284 (58.81)	7158 (0.04)	1575	3442	300	477	46.47
lymphA01	44,848,973	20,045,443 (44.70)	12,675,354 (63.23)	12,580,282 (62.76)	6081 (0.03)	2373	3651	300	421	48.74
spleen01	20,058,026	11,527,782 (57.47)	7,566,895 (65.64)	7,422,262 (64.39)	3161 (0.03)	945	1445	300	382	49.79
spleen02	35,359,899	15,350,468 (43.41)	9,888,746 (64.42)	9,739,169 (63.45)	7955 (0.05)	1857	6119	300	436	47.94
spleen03	34,350,848	19,055,973 (55.47)	11,356,082 (59.59)	11,244,710 (59.01)	5405 (0.03)	1194	4290	300	353	51.60
spleen04	42,861,276	19,038,817 (44.42)	12,498,406 (65.65)	12,394,988 (65.10)	7616 (0.04)	1442	4162	300	481	51.78
spleen08	37,544,029	15,975,904 (42.55)	10,761,939 (67.36)	10,516,975 (65.83)	7191 (0.05)	3516	5176	300	386	47.06
spleen11	35,405,980	15,273,939 (43.14)	9,877,753 (64.67)	9,726,051 (63.68)	6596 (0.04)	1351	5266	300	480	51.27
spleen12	21,926,554	12,590,769 (57.42)	8,383,040 (66.58)	8,298,012 (65.91)	5381 (0.04)	1298	989	300	415	58.16
spleen19	27,820,892	16,068,654 (57.76)	11,459,934 (71.32)	10,570,867 (65.79)	6288 (0.04)	6367	1553	300	381	42.60
Mean	36,082,370	16,728,724	10,723,361	10,317,004	11,123	3216	5486	300	460	50.32
Standard Deviation	9,831,029	3,477,218	2,487,225	2,214,377	15,630	2408	3348	0	67	4.11

Rm. rRNA clean: number and percentage of reads after removing ribosome sequence; Rm. Host clean: number and percentage of reads after removing host sequence; Virus reads: number of reads mapped to the virus database.

**Table 2 viruses-11-00979-t002:** Information of contigs with a high level of sequence similarity and then conformed as the Sendai virus. See table legend of Appendix A for detailed explanation of the table header.

Query ID	Subject ID	Identity (%)	Alignment Length	Mismatches	Gap Openings	q Start	q End	s Start	s End	e-Value	Bit Score
lung01|contig_245	AB005795.1	90.15	2072	204	0	1060	3131	4378	6449	0.0	2818
lung01|contig_302	AB005795.1	86.03	594	83	0	207	800	1451	2044	0.0	697
lung01|contig_307	DQ219803.1	90.06	513	50	1	22	534	9315	8804	0.0	691
lung01|contig_507	DQ219803.1	92.27	1747	135	0	99	1845	11,967	13,713	0.0	2542
lung01|contig_1156	DQ219803.1	87.60	500	62	0	99	598	7149	6650	1.5 × 10^−176^	623
lung01|contig_2161	AB005795.1	90.46	1362	130	0	101	1462	147	1508	0.0	1871
lung02|contig_1124	DQ219803.1	91.41	7028	602	2	1	7027	8306	15,332	0.0	9945
lung07|contig_34	DQ219803.1	91.44	596	51	0	1	596	5376	5971	0.0	845
lung07|contig_45	DQ219803.1	91.46	515	44	0	1	515	4576	4062	0.0	731
lung07|contig_444	DQ219803.1	89.01	1128	124	0	91	1218	2750	3877	0.0	1476
lung07|contig_506	DQ219803.1	90.68	794	74	0	2	795	976	183	0.0	1099
lung07|contig_798	DQ219803.1	92.43	1202	91	0	104	1305	14,131	15,332	0.0	1757
lung07|contig_1426	AB005795.1	91.60	607	51	0	131	737	10,163	9557	0.0	865
lung09|contig_2947	DQ219803.1	91.74	1550	122	4	1	1547	13,835	15,381	0.0	2206
lung11|contig_5506	DQ219803.1	91.31	656	57	0	72	727	5180	5835	0.0	926
lung19|contig_164	AB005795.1	89.76	13,231	1355	0	1	13,231	19	13,249	0.0	17,751

**Table 3 viruses-11-00979-t003:** Information of contigs with a high level of sequence similarity and then confirmed as Coronavirinae. See table legend of Appendix A for a detailed explanation of the table header.

Query ID	Subject ID	Identity (%)	Alignment Length	Mismatches	Gap Openings	q Start	q End	s Start	s End	e-Value	Bit Score	Taxonomy
lung07|contig_47	JX993987.1	80.24	506	100	0	2	507	7611	7106	2.28 × 10^−128^	462	Bat coronavirus Rp/Shaanxi2011
lung07|contig_174	KJ473815.1	87.16	1262	162	0	19	1280	15,069	16,330	0.0	1546	BtRs-BetaCoV/GX2013
lung07|contig_368	KC881006.1	88.93	1057	117	0	1	1057	28,204	29,260	0.0	1379	Bat SARS-like coronavirus Rs3367
lung07|contig_715	AY394981.1	88.68	521	59	0	96	616	17,937	17,417	0.0	673	SARS coronavirus HGZ8L1-A
lung07|contig_1748	DQ412042.1	87.84	584	71	0	46	629	11,919	12,502	0.0	733	Bat SARS CoV Rf1/2004
lung08|contig_223	KJ473814.1	85.52	2023	293	0	98	2120	14,509	12,487	0.0	2327	BtRs-BetaCoV/HuB2013
lung08|contig_286	KY417145.1	82.29	559	99	0	1	559	18,771	18,213	3.95 × 10^−158^	562	Bat SARS-like coronavirus
lung08|contig_330	FJ588686.1	80.46	1167	224	2	153	1317	1374	210	0.0	1072	SARS coronavirus Rs_672/2006
lung08|contig_424	DQ412042.1	87.99	608	73	0	3	610	12,054	12,661	0.0	767	Bat SARS CoV Rf1/2004
lung08|contig_729	KY417145.1	88.06	1139	136	0	1	1139	17,326	16,188	0.0	1442	Bat SARS-like coronavirus
lung08|contig_731	KF569996.1	83.96	767	123	0	2	768	11,978	11,212	0.0	829	Rhinolophus affinis coronavirus
lung08|contig_785	KF294457.1	83.45	1722	285	0	608	2329	21,463	19,742	0.0	1820	Bat SARS-like coronavirus
lung08|contig_1292	JX993988.1	82.61	644	112	0	3	646	24,133	24,776	0.0	657	Bat coronavirus Cp/Yunnan2011
lung08|contig_1420	AY394981.1	88.39	646	75	0	1	646	17,333	17,978	0.0	827	SARS coronavirus HGZ8L1-A
lung08|contig_1528	DQ412043.1	84.29	681	107	0	138	818	19,339	18,659	0.0	746	Bat SARS CoV Rm1/2004
lung08|contig_1551	GQ153548.1	82.60	500	87	0	2	501	24,202	23,703	1.73 × 10^−142^	509	Bat SARS coronavirus HKU3-13

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
