# Peer review of "Viral Metagenomics Revealed Sendai Virus and Coronavirus Infection of Malayan Pangolins (Manis javanica)"

_viruses, 2019, doi:10.3390/v11110979_

Round 1

Reviewer 1 Report

Manuscript No.:            viruses-617997

Title:                            Viral metagenomics revealed Sendai virus and coronavirus infection of Malayan Pangolins (Manis javanica)

Authors:                       Liu et al

In this study the authors used post-mortem tissue samples (lung, lymph, and spleen) to evaluate for the first time the virome of 11 Malayan Pangolins.  Among a series of viral sequences, they identified sequencing reads associated to coronaviruses and Sendai virus as the most prevalent in these dead animals.

Specific Comments:

Based on the Results described in this study, it is impossible to state that the 11 pangolins died as a consequence of infections by Sendai virus, coronavirus or any other viral infection. A more comprehensive study would need to be done, e.g., comparing the virome of healthy vs. ill/dead animals, recover viral particles (viruses) and not only viral sequences, etc.  All these statements need to be modified throughout the manuscript. Lines 76 to 80: It would be better to include some information about the library preparation and sequencing. Deep sequencing method and platform used?  For example, Briefly, ………

Minor Comments:

The English grammar should be revised.

Author Response

Comment 1: Based on the Results described in this study, it is impossible to state that the 11 pangolins died as a consequence of infections by Sendai virus, coronavirus or any other viral infection. A more comprehensive study would need to be done, e.g., comparing the virome of healthy vs. ill/dead animals, recover viral particles (viruses) and not only viral sequences, etc. All these statements need to be modified throughout the manuscript. Lines 76 to 80: It would be better to include some information about the library preparation and sequencing. Deep sequencing method and platform used? For example, Briefly, ………

Response: Many thanks for your advices, we completely agree with you that we can’t propose these pangolins died as a consequence of infections by Sendai virus, coronavirus or any other viral infection. In this paper, we just reported that we found these viruses from the dead ones. Meanwhile, it is also true as reviewer suggested that a more comprehensive study is needed to be done. However, as rare and protected animals, it is difficult to have the permission to obtain blood or tissue samples from healthy pangolin individuals. That is why we did not compare the virome of healthy and ill/dead animals. At the same time, just as reviewer suggested that recovering viral particles (viruses) is also important work, which we are still working on it, and we hope we will publish it soon.
Considering the reviewer’s suggestion of including some information about the library preparation and sequencing, we have re-written this part.

Minor Comments: The English grammar should be revised.

Response: The English grammar has been revised.

Special thanks to you for your good comments.

Reviewer 2 Report

The manuscript is very interesting and relevant. It focuses on the study of the unique and rare animal species Malayan pangolin (Manis javanica), which is endangered. The study aim is very specific and results from the urgent need to determine causes of the death of pangolins directly in the conditions of The Guangdong Wildlife Rescue Center and, even more, shed light on the danger of viral infections in these animals and possible pathways of viruses that enter the bodies of pangolins. The Authors use shotgun high-throughput sequencing, which has shown the efficiency in studying metagenomic samples of various types and species of macro- and microorganisms, including viral communities that inhabit different ecotopes. The Authors provide sufficient literature about the application of this method in the investigations of viromes of different animals and the results that were obtained owing to NGS.

I undoubtedly recommend the paper to be published and list only a few minor remarks below:

You should indicate why you chose mainly lungs for the study. Perhaps you suspected the presence of respiratory diseases by clinical signs in pangolins. I think that you should describe the symptoms of the disease and the histopathological pattern of organs from dead animals. What is the cause of their death according to veterinaries and pathologists? 
The lack of negative control reduces the reliability of the results. According to the Methods section, healthy or supposedly healthy animals were in Wildlife Rescue Center, i.e. not all animal became ill and died. In this case, it was necessary to take blood from them, make swabs from the mucous membranes of the nasal cavity and the trachea, or, if possible, even a lung biopsy. The Sendai virus can be found in the blood as well as in the muscle of the heart, liver and brain of sick animals.
Bacterial sequences most likely were present in viromes. If so, you should mention this as well as their percentage and names.
References lack no. 48, though this number is given in the text.
Figures 1, 2, 4 have a low resolution, and it is better to increase it.
 In Figure 1, you had better write "family" with a capital letter, and this picture can be increased in width of the sheet.
Table 1: There are no spaces between the title and the brackets. For example, сlean reads(PE) and beyond.
Why did you select the following values: Removal of reads with Ns more than 5% and read quality less than 20, and not, for example, 25 or 30?
Line 104: similarity≥ 90%. Add a space.
Line 168: monkey were downloaded from ViPR database: remove large spaces (highlight a large space, alt+ctrl+space).
Lines 224 - 227, 242-243: Change the font size.
Line 193: Genera - should start with a lowercase letter (?)

Author Response

Comment 1: You should indicate why you chose mainly lungs for the study. Perhaps you suspected the presence of respiratory diseases by clinical signs in pangolins. I think that you should describe the symptoms of the disease and the histopathological pattern of organs from dead animals. What is the cause of their death according to veterinaries and pathologists?

Response: We are very sorry for our negligence of describe the symptoms of the disease and the histopathological pattern of organs from dead pangolins. In fact, most of the dead pangolins have a swollen lung which contain frothy liquid, as well as the  symptom of pulmonary fibrosis. In minority of the dead ones, we observed hepatomegaly and splenomegaly. Therefore, we collected the tissues with observed symptoms to conduct the viral metagenomics study. We have added the related content in the text.

Comment 2: The lack of negative control reduces the reliability of the results. According to the Methods section, healthy or supposedly healthy animals were in Wildlife Rescue Center, i.e. not all animal became ill and died. In this case, it was necessary to take blood from them, make swabs from the mucous membranes of the nasal cavity and the trachea, or, if possible, even a lung biopsy. The Sendai virus can be found in the blood as well as in the muscle of the heart, liver and brain of sick animals.

Response: It is really true as reviewer suggested that negative control is very important. However, as rare and protected animals, it is difficult to have the permission to obtain blood or tissue samples from healthy pangolin individuals. At the same time, pangolins have a very poor ability to heal skin lesions, even a small wound may result in serious infection. We are afraid to run the risk of the death of these valuable individuals. Though it is very pity, we did not compare the virome of healthy and dead animals at this time, and this MS mainly reports the viruses found from these dead individuals. We will try to compare the virome of healthy and dead animals if we have samples in the future.

Comment3: Bacterial sequences most likely were present in viromes. If so, you should mention this as well as their percentage and names.

Response: It is really true that bacterial sequences were present in viromes, but in this paper we only focus on the viruses. We have a colleague who is working on the microbiome in the same samples as this paper. Meanwhile, we are still working on bacterial composition and found many kinds of bacterial species in these samples.

Comment 4: References lack no. 48, though this number is given in the text. 

Response: We are very sorry for our negligence, and we have added reference 48.

Comment 5: Figures 1, 2, 4 have a low resolution, and it is better to increase it.

Response: It is really true as reviewer said that these figures have a low resolution. We have made improvement according to the reviewer’s comments.

Comment 6: In Figure 1, you had better write "family" with a capital letter, and this picture can be increased in width of the sheet.

Response: We have made correction according to the reviewer’s comments.

Comment 7: Table 1: There are no spaces between the title and the brackets. For example, сlean reads(PE) and beyond.

Response: We have made correction according to the reviewer’s comments.

Comment8: Why did you select the following values: Removal of reads with Ns more than 5% and read quality less than 20, and not, for example, 25 or 30?

Response: The standard of quality control parameters we set is more stricter than the default parameters of SOAPnuke software. A paper published in Cell in 2018 just filtered out reads contained more than 30% low quality bases (Q≤2) or N bases (Liu et al., 2018. Genomic analyses from non-invasive prenatal testing reveal genetic associations, patterns of viral infections, and Chinese population history. Cell, 175, 347-359.). And we think our quality control standard could ensure we obtained high quality sequences. 

Comment9: Line 104: similarity≥ 90%. Add a space.

Response: We have made correction according to the reviewer’s comments.

Comment 10: Line 168: monkey were downloaded from ViPR database: remove large spaces (highlight a large space, alt+ctrl+space).

Response: We have made correction according to the reviewer’s comments.

Comment11: Lines 224 - 227, 242-243: Change the font size.

Response: We have made correction according to the reviewer’s comments.

Comment 12: Line 193: Genera - should start with a lowercase letter (?)

Response: We have made correction according to the reviewer’s comments.

Special thanks to you for your good comments.